# Evaluation of Affective Coexistence in Young Afro-Colombians in the Department of Chocó-Colombia

**DOI:** 10.3390/ijerph20021147

**Published:** 2023-01-09

**Authors:** Carolina Bringas Molleda, Manuel Beltrán Espitia, Yineth Mosquera Ruiz, Javier Herrero Díez, Francisco Javier Rodríguez Díaz

**Affiliations:** 1Department of Psychology and Anthropology, University of Extremadura, 10003 Cáceres, Spain; 2Department of Humanities, University Foundation Claretiana, Quibdó 270001, Colombia; 3Department of Psychology, University of Oviedo, 33003 Oviedo, Spain

**Keywords:** victimization, dating, Afro Colombians population, perception, age

## Abstract

Various works of research into violence in relationships between young couples refer to a lack of perception of some behavior patterns such as abuse. This means that the relationship has the potential risk of developing into one of victimization should it last into adulthood. Although it has been shown that this phenomenon may occur in any sector of the population, the interest of our study rests upon determining the prevalence of the perception of violent behavior patterns in relationships between adolescent and young adult couples. We also aim to analyze the differences obtained with respect to the characteristics of the aggressors in the young Afro-Colombian population of Quibdó, Colombia. The participants in the study consisted of 540 young Afro-Colombians of both sexes between 15 and 27 years of age. The instrument used was the reduced version of the Dating Violence Questionnaire. The results show a high level of victimization through violent behavior on the part of the partner, in great measure exercised by generalist aggressors. However, a small proportion could be perceived as abuse. The implications and possible means of intervention are discussed.

## 1. Introduction

The primary focus of this study is the research into violence in dating relationships during adolescence, being that this stage is when partnerships are forged for adult life and these interpersonal relationships at a young age also form individual habits for future affective relationships [1,2,3,4]. Violence in relationships between young people is still a very recent topic of investigation [5,6,7], and most of the research has focused on the violence in relationships where two young people are living together.

It is assumed that conduct is conformed by behaviours. And depending on the culture, these favor the adaptation of the individual to the medium in so far as these behaviours follow the rules that deem them as tolerable and/or give shape to a social attitude of inadaptation (illegal behaviours for adaptative goals vs formalization in psychopathological categories) [8,9]. The point of interest here is the violent, affective relationship between the aforementioned young people, as it is considered reciprocal and bidirectional; i.e., in such affective relationships, both members exercise violence on the other in one way or another [10,11]. Thus, the violent behavior of one of the partners can be a predictive factor of their own victimization [8].

It is assumed that this type of affective bond is not the same as that of domestic violence; the members do not live together, do not share their possessions, do not have legal formalities that tie them together as a couple, and do not have children in common [3]. Likewise, it is posited that one of the points of genesis of the violence in an affective adult interpersonal relationship is the interpersonal relationship formed by a pair young people, where behavioural patterns are already manifested as forerunners of future violence, including physical, verbal, emotional, or sexual abuse, among others [5,12,13]. Hereby, the risk assessment focused on discovering behavior patterns in affective relations between adolescents who have difficulty perceiving themselves as abusive. This poses a concrete challenge: to identify the protection and risk factors that, once detailed, can suppose a solid basis for orienting prevention programs.

Identifying violence in juvenile dating relationships tends to be difficult due to uncivil behavior patterns that are considered normal; this makes many people who are victims of violence (sexual, physical, or psychological, among others) incapable of recognizing the existence of that violence. This lack of perception is not new [14]; in fact, occurrence has been observed in adult women who suffered sexual harrassment [15]. Even though violent behavior patterns can be distinguished, the dissonance between labelling one as such and the experience of being abused opens several possibilities of grouping victims suffering from different types of violence. López-Cepero et al. (2015) [16] assume that this violence often occurs in relationships in which the woman is trapped or has suffered sexual violence; in such cases, there may be a dissonance between the experiences of the individuals suffering the abuse and their ability to allow themselves to be labelled as such. That is, although they are victims of the said violence, they are not capable of recognizing or accepting it.

The research that has been carried out over the last few years concerning violence in interpersonal dating relationships showed evidence that the increase in emotional and physical violence has consequences, even to the extent of altering behavior patterns which can result in the emergence of mental disorders. However, violence in dating adolescent couples has not received the same attention as that given to couples that live together. The violence suffered by adolescents and young adult couples is commonly of the physical type. This pattern is a concern due to its frequency of perpetration [17,18,19,20] and for being bidirectional; where women tend to initiate it to a greater extent than men [8,18]. It is also women who end up being more vulnerable than men; women tend to experience fear and serious injuries, even sexual abuse [5,13,21,22], with at least two types of aggressors being identified in young couples’ dating relationships: those that are violent exclusively and specifically with their current partner and those who, in fact, show this violence as one more sample or expression of their generalized violent behaviours in other environments or with other people. This is proposed, even though in the field of typology of aggressors, numerous studies include generalist aggressors in the classification [23,24,25,26,27] and/or reveal that variables such as sex, evaluation techniques, and the type of conduct shown by their peers constitute modulation factors of great importance, despite the aggressive behaviour of the couple, the anti-social and/or aggressive behaviour of their partners, and the victimization from peers being significantly related with the levels of perpetration and victimization of violence in adolescent dating relationships [8,28,29].

In addition to the consequences associated with the perpetration of the victimization in affective interpersonal dating relationships, the concern increases as the process starts to appear in the first stages of the life cycle, where affective relationships start (statistics refer to a greater presence of verbal violence, around 90%, than physical violence, almost 30%. The level is always higher in the adolescent population as opposed to couples with stable intimate relationships); these initial affective relationships are the ones that will forge the bonds in adult relationships [8,30]. It is in these moments when behavioral dynamics are consolidated, and therefore, serious consequences in establishing formal future relationships may be revealed [12,14,18,22].

It is known that a third of the juvenile population involved in dating relationships has experienced at least one relationship with violent episodes and/or sexual aggression, with psychological aggression being the type with the greatest prevalence. This type of violence can occur in different ways, stressing the one that takes place under pressure or without the victim’s prior consent [14,15,22].

Assuming that violence in couples can appear in any sector of the population, our interest lies in investigating the Afro-Colombian adolescent population and young adult couples residing in the department (an administrative subdivision similar to a state) of Chocó in Colombia. This population has serious difficulties associated with racial discrimination and governmental disregard, which push them to be part of the lowest strata and economical sectors in society, as well as to have conflictive interpersonal relationships in their social environment. This group is among the most vulnerable and poorest of the entire population. Consequently, the necessary conditions for a better quality of life are ever more limited due to restrictions with respect to the quality of education and labor market. Added to this, there are stereotypes concerning race; such stereotypes contribute to negative effects upon their ancestral and cultural practices, leading to restrictions on the performance of their rites and customs [31].

To understand dating violence in the Afro-Colombian population in the context of Chocó, it is important to know the demographics of this region. The local coordinating team in Chocó declares a population of 515,045, with a projection of 525,528 inhabitants for the year 2020 [32,33]. Data from the Governorship of Chocó (2017) [34] recognize different ethnic groups: Blacks or Afro-Colombians (75.68%), Native American Indians (11.9%), Mestizo (7.42%), and White (5.01%). DANE (2010) [33] has reported that more than 80% of the population residing in the department of Chocó identify themselves as “Mulato”, “Black”, “Afro-American”, or “Afro-Colombian”, with more than 12% belonging to the indigenous race and the remaining 7% corresponding to the Mestizo population of the department. Data from the State Prosecutor (2018) [35], which offers the guidelines on intrafamiliar violence, showed that during the 2017–2018 period, the evaluation of 191 cases of victims was carried out, showing an increase of 49.2% with respect to the figures for the 2016–2017 period, in which 17 protective measures were carried out. These findings represented an increase of over 13% from the previous period. The Public Defender’s Office (18th January 2018) [36] concerning feminicide and intrafamiliar violence on a municipal level reported that between January and August of 2017, there were four cases of feminicide; in the month of October, over 300 cases of intrafamiliar violence on investigation; and over 400 reports of victims of domestic violence in the same year.

These figures demonstrate the importance of the rate of violence against women in intimate relationships, which have implications for economic, physical, psychological, and sexual aggression. Taking into account all the particular conditions of the population of Chocó and its Afro-Colombian population, the social reality described provides the basis for us to examine the presence of mistreatment and the perception of violent behavioural patterns of abuse in the interpersonal affective relationships of dating in the juvenile Afro-Colombian population in Quibdó. Thus, our aim is:Determine the presence and perception of violent behavior patterns in dating relationships.Analyze the differences with respect to the characteristics of the aggressors in the juvenile Afro- Colombian population of Quibdó, Colombia.

## 2. Materials and Methods

### 2.1. Participants

The sample consisted of 540 young Afro-Colombians between the ages of 15 and 27 (*M* = 19.32; *SD* = 3.12). A total of 40.4% were men (*n* = 218; 40.4%), and almost 57% were studying or had completed university studies, while the rest were pursuing pre-university studies. All participants were single and did not live with their partners, with the time mean of their relationships being 19.63 months (*SD* = 22.10), and the age mean of their partners being 20 (*SD* = 5.30) A group of 348 participants (64.4%) had finished their secondary studies while the rest had completed the university level. At the same time, a total of 96 study participants (17.8%) had a job, while a total of 201 of their partners (37.2%) were currently working.

### 2.2. Measurement Instruments

The primary sets of variables were used for measures. First, a demographic form where participants were asked to respond to a series of questions about their age, gender, level of education, and job. They were also asked other questions concerning their previously referenced dating relationship, which prompted us to do the grouping for the analysis of the perception of abuse in the dating relationships of the teenage participants.

Second, the CUVINO-R, the short version of the Dating Violence Questionnaire [37], is made up of 20 behavioral items reflecting patterns of aggressive or abusive behavior towards a partner. This version comes from an original scale (CUVINO) by Rodríguez-Franco et al. (2010) [38] which contained 42 behavioral items, and was used between February and May 2019, being adapted to the African-American population according to the indications of Muñiz, Elosúa & Hambleton (2013) [39], and translated into English. The manner of responding to the 20 behavioral items is through a Likert-type scale from 0 (Never) to 4 (Almost always). The items are grouped into 5 factors of abuse: Detachment (i.e., item Does not acknowledge any responsibility regarding the relationship or what happens to both of you), Humiliation (i.e., item Criticizes you, underestimates the way you are, or humiliates your self-esteem), Sexual Abuse (i.e., item You feel compelled to have sex as long as you don’t have to explain why) Coercion (i.e., item “Tests” your love setting traps to find out if you are cheating), and Physical Abuse (For instance, item *Has beaten you*). Construct validity has been demonstrated by convergence with attitudes toward violence [40], gender roles attitudes [41], and labeling of dating experience [5]. In addition, the DVQ-R has a strong internal consistency with a Cronbach’s alpha estimate for the scores have ranged from 0.66 for Detachment and 0.84 for Coercion, and 0.87 as the reliability estimate for the total scores of the questionnaire [37].

Third, the participants were asked to answer three yes/no type questions about their perception of abuse and the type of aggressor they had faced as victims. The three questions were as follows: (a) Have you felt abused?; (b) Have you felt afraid of your partner?; (c) Has your partner been violent with anyone else (other friends, colleagues, etc.)? This procedure allowed for the establishment of three groups of analysis based upon the dichotomized variables abuse/nonabuse, generalized violence/exclusive violence in the couple: nonabused, abused by generalist aggressor, and abused by specialist aggressor.

### 2.3. Procedure

The data gathering was carried out through an invitation to participate in the study made to several high school and university institutions. Information about the objectives of the research was sent to each institution, as well as the criteria for the sample of participants: adolescents and young people of both genders who were involved in or had previously had a relationship of at least one month’s duration (similarly, the said relationship should not be a formal marriage with partners living together, but exclusively a dating relationship). The final sample includes the answers compiled from the educational institutions that accepted to participate in the study on victimization and perception of abuse in dating relationships. Participants were informed about the objectives of the study and informed about its anonymity and about the fact that the study was voluntary,; that participants were free to abandon the study at any moment for any reason. With intention to satisfy the ethical requirements regarding the inclusion of underaged participants, their explicit consent and the consent of their parents was requested, and clear information about the process was provided to the institutions. For those participants of legal age, consent was requested before the start of the evaluation. Anonymity was guaranteed by means of class group evaluation and results delivery only for complete samples. The researchers offered individualized information to address any possible discomforts or doubts related to the study.

The sample was divided based on two criteria: the perception of abuse in their affective relationships and the type of violence exerted by their partners. The grouping of participants by perception of abuse was done based on two criteria via combining the answers to two questions included in the questionnaire: «Do you feel or have felt abused by your partner?» and «Do you feel or have felt afraid in your affective relationship?» Thus, the subjects were assigned to the group of “nonabused” when there was a double negative in their answers, or to the group of “abused” when subjects were considered to have been abused or made to feel afraid during their couple’s relationship. Equally, not only participants with an awareness or perception of maintaining an abusive relationship were included but also those who were considered “technically abused”. That is to say, those who, even having provided evidence of being in an abusive relationship, did not have the perception nor the awareness of being abused. In this sense, the study follows the psychological considerations of previous research [3,18,40] that stress the need to include this population in this type of study.

The grouping of participants by Type of Violence is done by requesting each victim to indicate if their partner showed themself to be violent towards other people apart from their relationship. This allowed for the classification of the aggressors according to the specialization or generalization of their conduct. In other words, if their violent behaviours were oriented exclusively towards their partners or towards other people as well. This was proven through the question: «Does your partner shows itself violent with other people?» [3].

## 3. Variables and Data Analysis

First, we carried out the descriptive analyses of the participants’ own perception of abuse based on the two yes/no questions (Have you felt abused? Have you felt afraid of your partner?). Later, taking into account the third question referring to the possible violent behavior of the partner towards other people (Has your partner been violent with anyone else (other friends, colleagues, etc.?)), we obtained a variable that reflected the typology of abuse divided into three levels. The first level included those who answered negatively to the first two questions concerning their own perception of abuse, considering them as “They do not feel abused”. The second level refers to those who answered positively to one or both of the first two questions and also answered affirmatively to the third question; these cases were called “Abused by generalist aggressors”, as the partner was violent with other people. The third level refers to those who, as in the second group, answered positively to the first two questions but negatively to the third; in this case, they were denominated “Abused by specialist aggressors”.

In relation to the above and with the aim of discovering the differences between each of the factors of victimization with respect to the typology of abuse, we carried out the multivariate analysis of the variance (MANOVA), using the T3 test by Dunnet.

## 4. Results

Our aim is to determine the presence and perception of violent behavior patterns in dating relationships, as well as to analyze the differences with respect to the characteristics of the aggressors in the juvenile Afro-Colombian population of Quibdó, Colombia. In Table 1, the descriptive analysis of the victimization factors are shown, confirming that the detachment and coercion factors are the most frequent, as well as being the ones that offer a greater sample homogeinity, and thus, a greater representativeness (variation coefficients of 0.85 and 0.72, respectively). It is worthy of note that there are no missing values. Similarly, it is observed that the correlations between factors are statistically significant, with the correlations of detachment, humiliation and coercion factors showing a larger effect size (Table 2).

In Table 3, statistics are presented on the participants’ perception of abuse, in general. Approximately 11.3% (*N* = 61) of these young people reported being abused in their relationship, although 41% of those who perceived abuse did not report feeling afraid. On the other hand, from those who answered negatively to feeling abused (88.7%), some (19.8%) reported that, at some moment in their relationship, they had felt afraid of their partner. The differences obtained, χ^2^(1) = 45,214, *p* < 0.001, φ = 0.289, are statistically significant and offer us an effect size with an already important grade of implication for the relationship.

Table 4 shows the behavior patterns of victimization observed as answered by the participants. Thus, as for the general victimization suffered by our 15 to 27-year-old participants, as well as among the group that did not recognize themselves as having suffered abuse, three such behavior patterns exceeded the 40% threshold; these refer to Detachment, Coercion and Humiliation. Nevertheless, it can be seen that all five factors scored a high proportion (over 40%) among those who did recognize themselves as having suffered abuse or fear at any time (See Table 4).

Along the same lines, the typology of abuse shows us that a large proportion of our sample states that they did not feel abused or afraid at any moment in their relationship. However, among those who recognized themselves as feeling this way, 22.8% had been victimized by partners who were only violent with them and no one else, while only 6.1% had been aggressive with other people (See Table 5). Similarly, it was determined that 99.4% (*N* = 155) of those participants who referred to having lived through a situation of abuse had suffered two or more behavior patterns of victimization and 96.8% (*N* = 151) had suffered three or more. On the other hand, the data showed that among those who state that they had not experienced abuse or fear, 97.1% (*N* = 373) had experienced one or more behavior patterns of victimization by their partner, 88.5% (*N* = 340) had experienced two such behavior patterns and 78.6% (*N* = 302) had experienced three or more.

The results of the Multivariate Analysis of Variance showed each one of the factors of abuse compared with the others with respect to the typology of the aggressor, as well as their perception of victimization. Previously, the descriptive statistics demonstrated that for the five factors of victimization, the greatest proportion occurred in those participants who stated that their partner was also violent with other people (See Table 6).

Having proved that the typology of the perception of abuse offers significant differences (F = 11.458; Sig = 0.000) through the statistics of the Lambda of Wilks, the results of the intersubject tests can be seen in Table 7. These results showed statistically significant differences for all the factors of abuse. Furthermore, two of them showed a high level (values of 1) of Observed Power.

Finally, significant differences are shown from the results found for the five factors of victimization. These results allowed for the observation that those who did not perceive either abuse or fear during their relationship referred to themselves as having felt less victimization. The greatest victimization occurred among those young people who recognized and perceived themselves as having been abused by their partner; both those who only exercise violence against their partner and the generalist violent partners who also demonstrate their aggressive behavior beyond their affective relationships. If compared, on the other hand, it should be pointed out that the participants who stated they had felt abuse with both specialist and generalist aggressors showed significant differences for only one of the factors. To be precise, for the factor Detachment, those subjects who stated that their partners were generalist aggressors suffered more from this type of abuse than those whose partners restricted the violence to their partner alone (See Table 7).

## 5. Discussion

This research has been oriented towards determining the presence of abusive behavior patterns in the juvenile Afro-Colombian population in Quibdó, Colombia. The results show that the participants’ perception of abuse occurs in only a small proportion: only 11.3% reported feeling abused at some moment during their relationship. Nevertheless, this perception is not linked to a state of fearfulness of the partner; 41% recognized having felt afraid of their partner. By the same token, 88.7% of the studied population did not perceive abuse. However, 19.8% stated having felt afraid of their partner at some point in time during their dating relationship. These data coincide with the research of López-Cepero et al. (2015) [16] and Herrero-Olaizola et al. (2020) [8], who showed that the participants who perceived themselves as being abused were fewer in number than those who did not perceive themselves as being abused. However, of those who do not perceive abuse, a small percentage reported having felt afraid of their partner at some moment during their affective relationship, which is considered a result of unequal power relations. Therefore, it could be concluded that adolescents and young people learn coercive and aggressive behaviors towards their partner, as well as a negative communication style, due to the lack of behavioral models in their socialization [28,42]. In accordance with the above, it was found that adolescents who grew up in contexts in which they were witnesses or victims of violence were more likely to imitate or tolerate these behaviors in their relationships compared to adolescents who came from non-violent homes [43]. Similarly, Rey-Anacona (2015) [44] showed that teenagers and university students who exerted violence in their relationships more frequently reported having observed violence in their socialization. It is possible, according to these results, that the observation of models of violence generate attitudes which favor the legitimization of the use of partner abuse in adolescents and young people [30,45].

As for the behavior patterns of victimization, it was found that the young participants, even when they did not perceive abuse in their affective dating relationships, were victims of violent behavior patterns concerning mainly patterns such as Detachment, Coercion and Humiliation. This coincides with the proportion of young Afro-Colombians who suffered general victimization. The fact that some of those who did not perceive abuse still recognized having experienced more than one form of violent behavior by their partner continues to draw attention.

These results repeat and coincide with the research carried out by Cortés-Ayala et al. (2015) [18] and Rodríguez-Díaz et al. (2014) [5]. Of these, it should be pointed out that the studies were carried out using different populations and at different times. The results are consistent with the perception of those young people who acknowledged having suffered abusive situations; they are the ones who had experienced the most violent behavior by their partner. According to the research carried out by Vivanco et al. (2015) [20], as well as that carried out by Herrero et al. (2020) [8], the results reflect the experience of some form of violence within the couple, either as a victim or as an aggressor; unidirectionality is considered to be substantially lower, while mutual aggression offers higher levels of both aggression and victimization. This implies the need to consider all the dimensions present in both aggression and victimization in dating relationships. In our study, the majority of the young people affirmed having suffered more than one form of violent behavior on the part of the partner, which, following the results of Herrero et al. (2020) [8], would be within the upward bias of victimization scores. That is, the study participants who acknowledged being aggressive with their partner would tend to show higher victimization scores.

The typologies of abuse, and with respect to the characteristics of the possible aggressors, offer results that showed an ample proportion of the population under study referred to not feeling abused, nor had they felt fear in their dating relationship. Of those subjects who did perceive abuse, 22.6% had been victimized by their partner, who was only violent within the relationship and not outside it; only 6.1% referred to having been abused by a partner who, at some moment, was also violent towards other people. Nevertheless, at the time of establishing the comparison with respect to each one of the factors of victimization, it was observable that, as in the results obtained by Rodríguez-Franco et al. (2017) [3], the generalist aggressors were the ones who exercised greater violence in all the factors studied when compared to the specialist aggressors (while those who referred to not having suffered abuse or fear were even lower). To be precise, and comparing the type of aggressor in our study, the differences were found in victimization due to the factor of Detachment, i.e., those subjects whose partner was a generalist aggressor suffered this type of abuse to a greater extent than those whose partner was violent solely with them. Nevertheless, these results do not agree with the research carried out by Rodríguez-Franco et al. (2017) [3], since, in their results, this factor did not reflect differences between both types of aggressor; in fact, the differences appeared in abuse due to humiliation and coercion. However, these findings in our study can be the result, unlike in theirs, of having used the abbreviated version of the instrument (i.e., five factors of abuse instead of eight in the original version).

### Limitations

The present investigation is not free of potential limitations. First, the participants belonged to the general population; perhaps, for this reason, the levels of aggression and victimization found were low in any case, so a generalization to other risk groups (couples with more serious aggressions, for example) must be done with a great deal of caution. Similarly, our research has allowed us to see, through the cultural adaptation of the questionnaire, the difficulty of understanding the item: Do you not recognize your responsibility in the relationship? The participants interpreted this as meaning at the moment of giving their reply, rather than as a behavioral action present in the daily affective relationship. We aim to improve the aforementioned item for this population context in our future research. Similarly, it is necessary to carry out a differential analysis of the incidence in the obtained results of the following variables: duration of the relationship, degree of commitment and/or maturity of the people in the affective dating relationships. In accordance with the results, the said differential analysis should control the academic educational level, so as to allow for the extrapolation of the results to the entire Afro-American population under study; in other words, dating violence in our research was not modeled at the situation or event level (that marked cultural differences, as well as economical status, have been noted in the aformentioned context), and the data at the partner level come from a single self-report measure, which may not easily reflect specific incidents of serious gender-based violence. It is also important to include the analysis of violence that occurs in an online context through the use of social networks in future studies, as these mediums could be used to intimidate a partnerthrough various manifestations. Finally, the cross-sectional nature of the study, at a particular moment in time and concerning a concrete relationship, seems to indicate the need to go deeper into their relationships and the resolution of problems that appear in the dating couples; the cross-sectional nature of the data does not allow the antecedents of the consequences to be distinguished, limiting the generalizability of the results.

These limitations of our results will hopefully be overcome in later studies, so as to be able to set a framework for and orient preventive interventions that can be developed in the Afro-Colombian context, where, to establish the prevalence of these behaviors, the convenience sampling should be changed to a random one and consider the study of couples.

## 6. Conclusions

The results, therefore, refer to the need to respond to the discrepancy between the aggression and victimization scores. The study of couples offers numerous advantages for this, although it also reveals limitations that are opaque when exclusively analyzing one member of the couple, at least as long as the multiple sources of bias that may be interfering in the responses to self-reports are not clarified.

## Figures and Tables

**Table 1 ijerph-20-01147-t001:** Descriptive analysis of the victimization factors.

Factors of Victimization	*M*	*SD*	CV	Range	Skewness	Kurtosis
Detachment	2.92	2.49	0.85	15	1.19	1.99
Humiliation	0.98	1.31	1.33	7	1.79	3.16
Sexual	0.76	1.43	1.88	11	2.83	10.46
Coercion	3.11	2.24	0.72	13	0.88	0.94
Physical	0.60	1.28	2.13	12	3.73	19.70

**Table 2 ijerph-20-01147-t002:** Analysis of correlations and effect size.

Factors of Victimization	Detachment	Humiliation	Sexual	Coercion
Humiliation	0.283 ** (0.080)			
Sexual	0.234 ** (0.054)	0.179 ** (0.032)		
Coercion	0.283 ** (0.080)	0.194 ** (0.037)	0.265 ** (0.070)	
Physical	0.187 ** (0.034)	0.211 ** (0.44)	0.239 ** (0.057)	0.250 ** (0.062)

** *p* < 0.01.

**Table 3 ijerph-20-01147-t003:** Self-perception of abuse in young people of both sexes between 15 and 27 years old.

		Abuse		Total
		Yes	No	
Fear	Yes	36 (59%)	95 (19.8%)	131 (24.3%)
	No	25 (41%)	384 (80.2%)	409 (75.7%)
Total		61 (100%)	479 (100%)	540 (100%)

**Table 4 ijerph-20-01147-t004:** Factors of victimization suffered by young people between 15 to 27 years old.

Factors of Abuse	General Victimization		Abuse Not Perceived		Abused	
	*N*	%	*N*	%	*N*	%
Detachment	452	83.7	306	79.7	146	93.6
Humiliation	292	54.1	184	47.9	108	69.2
Sexual	193	35.7	110	28.6	83	53.2
Coercion	490	90.7	339	88.3	151	96.8
Physical	169	31.3	90	23.4	79	50.6

**Table 5 ijerph-20-01147-t005:** Typology of self-perception of abuse.

Abuse	
Do not feel abused	384 (71.1%)
Abused by specialist aggressors	123 (22.8%)
Abused by generalist aggressors	33 (6.1%)

**Table 6 ijerph-20-01147-t006:** Descriptive analyses of the factors of abuse with respect to the self-perception of abuse.

Factors	Typology of Abuse	Average	Standard Deviation
Detachment	Do not feel abuse	2.53	2.27
	Abused by specialist aggressors	3.49	2.58
	Abused by generalist aggressors	5.24	2.98
Humiliation	Do not feel abuse	0.82	1.20
	Abused by specialist aggressors	1.26	1.37
	Abused by generalist aggressors	1.81	1.70
Sexual	Do not feel abuse	0.54	1.23
	Abused by specialist aggressors	1.09	1.53
	Abused by generalist aggressors	1.96	2.15
Coercion	Do not feel abuse	2.84	2.12
	Abused by specialist aggressors	3.68	2.44
	Abused by generalist aggressors	4.03	2.28
Physical	Do not feel abuse	0.36	0.86
	Abused by specialist aggressors	1.04	1.73
	Abused by generalist aggressors	1.69	2.21

**Table 7 ijerph-20-01147-t007:** Analysis of MANOVA on the differential effect of the typology of the self-perception of abuse in each of the considered factors of victimization.

Factors			Dif. Av		
	F GL	P Partial Eta Squared	NA-SA Dif. Av. g	NA-GA Dif. Av. g	GA-SA Dif. Av. g
Detachment	23.948 2/537	<0.001 0.082	−0.95 * 0.409 (0.23–0.58)	−2.70 * 1.162 (0.856–1.469)	1.74 * 0.665 (0.328–0.984)
Humiliation	13.124 2/537	<0.001 0.047	−0.44 * 0.354 (0.18–0.52)	−0.99 * 0.795 (0.493–1.097)	0.54 0.381 (0.056–0.705)
Sexual	20.782 2/537	<0.001 0.072	−0.54 * 0.420 (0.248–0.592)	−1.42 * 1.073 (0.768–1.377)	0.87 0.119 (0.192–0.845)
Coercion	9.651 2/537	<0.001 0.035	−0.83 * 0.382 (0.21–0.553)	−1.18 * 0.558 (0.258–0.858)	0.34 0.195 (−0.177–0.408)
Physical	28.109 2/537	<0.001 0.095	−0.67 * 0.600 (0.427–0.773)	−1.32 * 1.292 (0.985–1.600)	0.64 0.353 (0.029–0.677)

* <0.05. LW: Lambda de Wilks; NA: Feel no abuse; SA: Abuse by specialist aggressors; GA: Abuse by generalist aggressors; Dif. Av.: Differences in averages; g: Size of Hedges effect. [LW = 0.815, GL = 10/1066, *p* < 0.001, eta *p* = 0.097].

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
