# Peer review of "Evaluation of Affective Coexistence in Young Afro-Colombians in the Department of Chocó-Colombia"

_ijerph, 2023, doi:10.3390/ijerph20021147_

Round 1

Reviewer 1 Report

The article offers an interesting food for thought on dating violence in the population of adolescents and young adults, in particular it poses an important question on the recognition of the violence of violent attitudes suffered and the perception of the aggressor. In order to improve the reading of the article I suggest for the authors:

Section introduction 

1) Avoid using the term young to refer to both populations because they are two different moments of life and with different characteristics (adolescents and young adults), so I would suggest writing throughout the text dating violence in adolescent and in young adult, as  it is reported only in some parts of the manuscript;

2) Add references in the literature on dating violence in adolescents and young adults; and

3) Add if there is data on dating violence in adolescents and young adults. At the end of the introduction, the authors inserted  data on violence in intimate relationships but they did not detail on this specific age group.  there is no data about it?

Section Measurement Instruments

The authors should  explain better the following  passage, because I have not understood how is composed the instrument: this measure was put together from the data provided by an original scale 157 version (CUVINO) by Rodríguez-Franco et al. (2010) [39], which contained 42 behavioral 158 items. The new version is made up of 20 behavioral items reflecting patterns of aggressive 159 or abusive behavior towards a partner.

Section Procedure

I find it confusing. In the procedures subsection the authors insert the division of the interviewees with respect to the two questions asked (« Do you feel or have felt abused 200 by your partner?» and « Do you feel or have felt afraid in your affective relationship?»). I propose to move it to the results section, because it is a detected data on which the other variables of the questionnaire are then considered.

A curiosity are there differences between adolescents and young adults? 

I suggest also to report as a limit (that could be a future study) that the authors did not explicitly refer to the contexts of dating violence, distinguishing between context online and the off-line where forms of violence have been suffered.

Author Response

We would like to thank our study for the review. This has provided us with suggestions for improving the article, so we now respond to the recommendations made by the reviewers.

Section introduction 

1) Avoid using the term young to refer to both populations because they are two different moments of life and with different characteristics (adolescents and young adults), so I would suggest writing throughout the text dating violence in adolescent and in young adult, as  it is reported only in some parts of the manuscript

Response: Our intention is to refer to violence in young couples, who have not started living together. We distinguish two age strata: Adolescents and young adults. In order not to be repetitive, we refer to both groups globally. However, it has been clarified with the reviewer's proposed terms (adolescents and young adults) in some sections of the text.

2) Add references in the literature on dating violence in adolescents and young adults

Response: The article contains continuous references to studies that deal with violence in adolescent and young adult couples. Some of them in recent years. A good example of this are reference numbers 1, 2, 5, 6, 7, 11, 12, 13, 14, 16, 17, 18, 19, 20, 22, 28, 30, 37, 39, 40, 41, 42, 44

3) Add if there is data on dating violence in adolescents and young adults. At the end of the introduction, the authors inserted  data on violence in intimate relationships but they did not detail on this specific age group.  there is no data about it?

Response: The authors have carried out a review of the literature on violence in young couple relationships (adolescent and young adults) within the theoretical framework. However, there is little quality research published on this violence in the youth of Chocó (Colombia), at least in these age groups. That is why we end our theoretical section with a summary of statistical data on violence in this Afro-American population, based on studies offered by DANE, the Chocó government, the Attorney General's Office, or the Ombudsman's Office. This serves as justification for carrying out our study in the younger population, which allows us to know the prevalence of cases of violence, as well as related variables.

Section Measurement Instruments

Reviewer: The authors should  explain better the following  passage, because I have not understood how is composed the instrument: this measure was put together from the data provided by an original scale version (CUVINO) by Rodríguez-Franco et al. (2010), which contained 42 behavioral items. The new version is made up of 20 behavioral items reflecting patterns of aggressive or abusive behavior towards a partner.

Response: The paragraph has been clarified, ordering the explanation

Section Procedure

Reviewer: I find it confusing. In the procedures subsection the authors insert the division of the interviewees with respect to the two questions asked (« Do you feel or have felt abused by your partner?» and « Do you feel or have felt afraid in your affective relationship?»). I propose to move it to the results section, because it is a detected data on which the other variables of the questionnaire are then considered.

Response: These dichotomous questions are explained in the "procedure" section because it refers to one of the categorization criteria of the participating sample. These are not results in the strict sense, but rather it explains the classification that is made of the sample according to a variable of the participants (feeling of fear or abuse), to then explain the statistical analyzes used and the results that are expected to be achieved.

Reviewer: A curiosity are there differences between adolescents and young adults? 

Response: The objective of this study is not to analyze the differences by age, but rather to assess the prevalence of violence in young, unmarried couples who have not started cohabiting. However, there are some statistically significant differences by age segmentation in some of the victimization factors. We reiterate, however, that this is not the objective of our research and that we hope to answer it in future studies.

Reviewer: I suggest also to report as a limit (that could be a future study) that the authors did not explicitly refer to the contexts of dating violence, distinguishing between context online and the off-line where forms of violence have been suffered.

Response: Our study has really dealt with dating violence, in a face-to-face context. It is true that the forms of violence have increased in recent years, spreading through the online modality (social networks). We are aware of this and it leads us to consider this reality as an important possibility; At the same time, it is presented as one of the limitations to be answered in subsequent studies.

Reviewer 2 Report

The study sets out to analyze when abuse/violence is perceived among young couples. The overall results showed that the prevalence of aggressive behaviors was generally high, but these behaviors were commonly not perceived as abuse. This has extremely significant implications for programs/policies related to what is perceived as abuse, what is legally considered abuse, and what can be done effectively reduce abuse whether perceived or otherwise among young couples. 

Excellent study overall, very interesting and significant. Just give the manuscript a thorough proofread for grammar. 

Author Response

We would like to thank our study for the review. This has provided us with suggestions for improving the article, so we now respond to the recommendations made by the reviewers.

Reviewer 2

The study sets out to analyze when abuse/violence is perceived among young couples. The overall results showed that the prevalence of aggressive behaviors was generally high, but these behaviors were commonly not perceived as abuse. This has extremely significant implications for programs/policies related to what is perceived as abuse, what is legally considered abuse, and what can be done effectively reduce abuse whether perceived or otherwise among young couples. 

Excellent study overall, very interesting and significant. Just give the manuscript a thorough proofread for grammar. 

Response: The article has been revised, using a native translator. Despite this, if it were considered necessary for its publication, we would be willing to follow your instructions.

Round 2

Reviewer 1 Report

I have no comments for the authors